# Site Selectivity in Pd-Catalyzed Reactions of α-Diazo-α-(methoxycarbonyl)acetamides: Effects of Catalysts and Substrate Substitution in the Synthesis of Oxindoles and β-Lactams

**DOI:** 10.3390/molecules24193551

**Published:** 2019-09-30

**Authors:** Daniel Solé, Ferran Pérez-Janer, Arianna Amenta, M.-Lluïsa Bennasar, Israel Fernández

**Affiliations:** 1Laboratori de Química Orgànica, Facultat de Farmàcia i Ciències de l’Alimentació, Universitat de Barcelona, Av. Joan XXIII 27-31, 08028 Barcelona, Spain; ferranperez1@hotmail.com (F.P.-J.); amenta.arianna@gmail.com (A.A.); bennasar@ub.edu (M.-L.B.); 2Departamento de Química Orgánica I and Centro de Innovación en Química Avanzada (ORFEO-CINQA), Facultad de Ciencias Químicas, Universidad Complutense de Madrid, 28040 Madrid, Spain

**Keywords:** palladium, diazo compounds, carbenes, oxindoles, β-Lactams, density functional calculations

## Abstract

The Pd-catalyzed intramolecular carbene C–H insertion of α-diazo-α-(methoxycarbonyl)acetamides to prepare oxindoles as well as β-lactams was studied. In order to identify what factors influence the selectivity of the processes, we explored how the reactions are affected by the catalyst type, using two oxidation states of Pd and a variety of ligands. It was found that, in the synthesis of oxindoles, ((IMes)Pd(NQ))_2_ can be used as an alternative to Pd_2_(dba)_3_ to catalyze the carbene C_Ar_sp^2^–H insertion, although it was less versatile. On the other hand, it was demonstrated that the Csp^3^–H insertion leading to β-lactams can be effectively promoted by both Pd(0) and Pd(II) catalysts, the latter being most efficient. Insight into the reaction mechanisms involved in these transformations was provided by DFT calculations.

## 1. Introduction

In recent years, the development of new methodologies for the selective functionalization of unactivated C–H bonds has become a very active area of research [1,2]. As part of this exciting field, the transition metal-catalyzed intramolecular carbene C–H insertion by decomposition of α-diazocarbonyl compounds has emerged as a powerful methodology for the construction of carbocyclic and heterocyclic frameworks [3,4,5,6,7,8]. A number of transition metal complexes have been used as effective catalysts to generate reactive metallacarbenes starting from α-diazocarbonyl compounds [9,10,11,12,13,14,15,16]. Among them, rhodium(II) [17,18], copper(I) [19,20], and more recently ruthenium(II) catalysts [21,22,23,24,25,26,27] have been proven to be especially useful for the development of highly selective carbene C–H insertion methodologies via a variety of reaction modes. Interestingly, palladium, one of the most commonly employed metals in homogeneous catalysis, remains underexploited in this type of carbene C–H insertion processes. Thus, while the effectiveness of palladium complexes in catalyzing carbene C–H insertion reactions from α-diazo carbonyl compounds was demonstrated some time ago [28], their use has been restricted to a couple of examples of insertion into C_Ar_sp^2^–H bonds [29,30,31]. This fact is highly surprising if we take into account the great success of palladium catalysis in cross-coupling reactions of diazo compounds with organic halides, pseudohalides, or arylboronic acids [32,33,34,35] and that the palladium-catalyzed cyclopropanation of olefins with diazomethane is a widely used synthetic methodology [36].

The longstanding research on transition metal-catalyzed carbene C–H insertion has generated an extensive literature on the use of dirhodium(II) catalysts to promote the intramolecular C–H insertion of α-diazoacetamides [37,38], as the insertion products, namely β- and γ-lactams as well as 2-oxindoles, are common scaffolds found in numerous natural products. In recent years, some ruthenium(II) catalysts have also been applied to promote this kind of C–H insertion process [21,22,25,39,40,41,42,43]. It has been shown that the site selectivity of these reactions not only depends on the type of diazocarbonyl compound but also is governed by conformational, steric, as well as electronic factors [44,45,46]. Moreover, some dramatic ligand effects have also been observed. For example, the use of carboxylate and, in particular, carboxamide ligands in dirhodium(II) catalysts has resulted in highly chemo-, regio- and stereoselective transformations [47,48] despite a variety of potentially competitive carbene-mediated processes (Scheme 1).

As part of our research program on the synthesis of nitrogen heterocycles, we have been exploring different ways to increase the versatility of palladium catalysis in C–C bond-forming reactions [49,50,51], for example, by controlling the ambiphilic character of the organopalladium intermediates in the intramolecular coupling reactions with carbonyl derivatives [52]. Continuing our search for methodologies to enhance the synthetic potential of organopalladium chemistry, we have also investigated the versatility of palladium as a catalyst for the carbene C–H insertion. In this context, we reported that palladium catalysts are able to promote Csp^3^–H insertion of carbenes derived from α-diazoesters to form pyrrolidines through intramolecular Csp^3^–Csp^3^ bond formation [53,54].

We have also demonstrated that the carbene C_Ar_sp^2^–H functionalization of α-diazo-α-(methoxycarbonyl)acetanilides to give oxindoles can also be promoted by using Pd_2_(dba)_3_ as the catalyst [55]. This allowed us to develop a one-pot methodology to prepare 3-(chloroethyl)oxindoles by means of a sequential C–H insertion/alkylation process (Equation (a) in Scheme 2). More recently, we have described the synthesis of β-lactams by Pd(II)-catalyzed carbene Csp^3^–H insertion of α-diazo-α-(methoxycarbonyl)acetamides (Equation (b) in Scheme 2) [56].

The aim of the current work was to gain more insight into the Pd-catalyzed carbene insertion reactions of α-diazo-α-(methoxycarbonyl)acetamides leading to oxindoles and β-lactams. In order to ascertain the factors governing the selectivity of the processes, we explored how the reactions are affected by the substituents on α-diazoacetamide and the type of catalyst, using complexes with two oxidation states of Pd and a variety of ligands (Scheme 3). Herein, we present a full account of our experimental and computational studies on the intramolecular insertion of α-diazo-α-(methoxycarbonyl)acetamides using Pd(0) and Pd(II) complexes, focused on identifying differences in the catalyst reactivities and selectivities.

## 2. Results and Discussion

The oxindole system is a common structural motif in pharmaceuticals and natural products, and the development of efficient procedures for their preparation is, therefore, of considerable interest [57,58,59,60].

Our previous studies on the carbene reactions of α-diazo-α-(methoxycarbonyl)acetanilides showed that, when using palladium catalysts, the C–H insertion giving the oxindole occurs selectively in the C_Ar_sp^2^–H rather than the Csp^3^–H bonds (Equation (a) of Scheme 2) [55]. The optimization process with α-diazoacetanilide **1a** revealed that the carbene C_Ar_sp^2^–H insertion can be selectively promoted by both Pd(0) and Pd(II), the best result being obtained when using Pd_2_(dba)_3_ in the presence of Cs_2_CO_3_ in dichloroethane at reflux for 96 h, which afforded oxindole **2a** in 66% yield (Table 1, entry 1). However, this reaction required a high catalyst loading and long reaction time. All attempts to increase the efficiency of the Pd(0)-catalyzed reaction by adding different phosphine ligands failed, always resulting in a slower reaction rate.

In order to improve the efficiency of the carbene insertion of α-diazoacetanilide **1a**, in the present study, we tested some palladium catalysts bearing non-phosphine ligands. (Pd(allyl)Cl)_2_ was first explored for the insertion/alkylation sequential process (Table 1, entries 2–3), but significant amounts of the starting material were recovered even after 60 h. In contrast, the use of ((IMes)Pd(NQ))_2_ required a notably shorter reaction time and lower catalyst loading (Table 1, entry 4).

With these data in hand, we then explored the generality of the NHC-Pd(0) catalyst in the sequential C–H insertion/alkylation process, leading to 3-(chloroethyl)-3-(methoxycarbonyl)oxindoles. Table 2 gathers the results of the reactions of different α-diazoacetanilides (**1b**–**1j**) when using ((IMes)Pd(NQ))_2_ as the catalyst and compares them with those previously obtained with Pd_2_(dba)_3_ [55].

As can be seen in Table 2, like Pd_2_(dba)_3_, ((IMes)Pd(NQ))_2_ also selectively promoted the carbene insertion into the arylic C–H bond in substrates having substituents at the nitrogen atom with primary, secondary, as well as tertiary Csp^3^–H bonds. Notably, ((IMes)Pd(NQ))_2_ was more efficient than Pd_2_(dba)_3_ in promoting the insertion of *N*-alkylacetanilides lacking substituents on the arylic ring (**1b**–**1d**) (Table 2, entries 1–6) or bearing an electron-donating substituent (**1g**) (Table 2, entries 11–12). In contrast, the introduction of electron-withdrawing groups dramatically diminished the catalytic efficiency of ((IMes)Pd(NQ))_2_. Thus, for example, acetanilide **1f**, bearing an *ortho*-bromo substituent, was recovered unchanged when using ((IMes)Pd(NQ))_2_, whereas in the presence of Pd_2_(dba)_3_, it afforded oxindole **2f** in 23% yield (Table 2, entries 9 and 10). Oxindole **2h**, with a fluoro group, was isolated in a modest 39% yield when using ((IMes)Pd(NQ))_2_ but obtained in an acceptable 64% yield in the presence of Pd_2_(dba)_3_ (Table 2, entries 13–14). No insertion product was formed in the ((IMes)Pd(NQ))_2_-catalyzed reaction of acetanilide **1i**, which bears a methoxycarbonyl substituent, while oxindole **2i** was isolated in 22% yield when using Pd_2_(dba)_3_ (Table 2, entries 15–16). ((IMes)Pd(NQ))_2_ also gave worse results in the sequential insertion/alkylation sequence from *N*,*N*-diphenylacetamide **1e** (Table 2, entries 7 and 8) and *N*-naphthylacetamide **1j** (Table 2, entries 17–18).

At this point, we studied the decomposition of α-diazo-*N*-pyridinylacetamide **3**, which, upon treatment with a catalytic amount of ((IMes)Pd(NQ))_2_ under the optimized reaction conditions, afforded mesoionic compound **4** (77%), resulting from the interception of the transient metallacarbene by the pyridine nitrogen (Scheme 4). This kind of mesoionic imidazopyridines have been previously prepared [61] and show interesting insecticidal activity [62].

Regarding the palladium-catalyzed reactions shown in Table 1 and Table 2, it should be noted that the formation of β-lactam products, resulting from the possible competitive carbene Csp^3^–H insertion, was not observed. This sharply contrasts with the competition between the carbene C_Ar_sp^2^–H and Csp^3^–H insertions observed in Rh(II)-promoted processes (see, for instance, Scheme 1(a)) [48].

Our previously reported DFT calculations suggested that the Pd(0)-catalyzed C_Ar_sp^2^–H insertion of α-diazo-α-(methoxycarbonyl)acetanilides to give oxindoles proceeds via a genuine stepwise mechanism involving a Pd-mediated 1,5-hydrogen migration from the initially generated pallada(0)carbene complex, followed by a reductive elimination [55]. It was also shown that the complete chemoselectivity of the process, which exclusively produces oxindoles over β-lactams, takes place mainly under kinetic control. We now decided to check the generality of this unprecedented Pd(0)-mediated mechanism using the model Pd(NHC) (NHC = 1,3-bis(phenyl)-imidazil-2-ylidene) catalyst.

Figure 1 shows the computed reaction profile for the formation of oxindole **2b’**, precursor of the experimentally observed oxindole **2b** (see Table 2). As expected, based on our previous report [55], the process begins from pallada(0)carbene **INT0**, formed upon reaction of α-diazoamide **1b** and the active catalyst species Pd(NHC). This species evolves into intermediate **INT1** in a highly exergonic process (∆G_R_ = –25.3 kcal/mol) through the transition state **TS1** with a computed activation barrier of 16.2 kcal/mol, which is fully compatible with the reaction conditions used in the experiment. Closer inspection of **TS1** indicates that this reaction step can be viewed as a Pd-mediated 1,5-hydrogen migration, which involves the formal oxidation of the transition metal. The alternative Csp^3^–H activation reaction via **TS1’**, which would involve an analogous Pd-mediated 1,4-hydrogen migration, is not competitive in view of the much higher activation barrier (∆G^≠^ = 8.2 kcal/mol) computed for this alternative transformation. Therefore, it is once again found that the complete selectivity of the process takes place under kinetic control. The transformation ends with an exergonic (∆G_R_ = –6.8 kcal/mol) reductive elimination reaction that directly transforms **INT1** into oxindole **2b’** with the concomitant release of the active catalytic species Pd(NHC) through the transition state **TS2** (with a feasible activation barrier of 18.7 kcal/mol).

On the other hand, we have also reported that the intramolecular carbene C–H insertion of α-diazo-α-(methoxycarbonyl)acetamides to form β-lactams can be effectively catalyzed by Pd(II) complexes (Scheme 2b) [56]. Due to its ubiquitous presence in the molecular structure of natural products and biologically active compounds, β-lactam synthesis has attracted considerable attention over the years [63,64,65]. To further understand the impact of the electronic nature of the catalyst in the aforementioned insertion reaction, we decided to explore the use of some Pd(0)-precatalysts. Table 3 shows the results of the reactions of diversely substituted *N*-benzyl-*N*-*^t^*butyl-α-diazoacetamides (**5a**–**5m**) when using either Pd_2_(dba)_3_ or ((IMes)Pd(NQ))_2_ as the catalyst and compares them with those previously obtained in the Pd(II)-catalyzed reactions [56].

The examples in Table 3 confirm the generality and functional group tolerance of Pd_2_(dba)_3_ and ((IMes)Pd(NQ))_2_ as catalysts for these insertion processes. Although both Pd(0) and Pd(II) catalysts can be used to promote Csp^3^–H insertion, some significant differences were identified. On the whole, the Pd(II) complexes proved to be more versatile for β-lactam formation despite not always giving the highest yield. In general, the resulting β-lactams were obtained in moderate to good overall yields (53–90%), usually as mixtures of *cis* and *trans* isomers, in transformations proceeding with high site selectivity, except when using **5f**. In this particular case, the use of Pd(II) catalysts to promote its decomposition resulted in the formation of *trans***-7f** (10–14%), together with major amounts of aldehyde **10**. On the other hand, when ((IMes)Pd(NQ))_2_ was used, a complex reaction mixture was obtained, from which *cis*-**7f** (9%), *trans*-**7f** (10%), and aldehyde **10** (15%) were isolated, while no cyclization product was obtained in the presence of Pd_2_(dba)_3_.

Notably, partial or complete isomerization of the *cis* β-lactam to the more stable *trans* isomer took place during the chromatographic purification of the reaction mixtures, which explains the observed differences in *cis*/*trans* ratios before and after purification.

The effect of adding phosphine ligands to the Pd(0) catalyst was also explored. However, similarly to the oxindole-forming reactions, the use of Pd_2_(dba)_3_ in the presence of phosphine ligands resulted in slower reaction rates. Thus, for example, treatment of **5a** with Pd_2_(dba)_3_ (10 mol%) in the presence of (*o*-tolyl)_3_P (20 mol%) under otherwise the same reaction conditions gave a mixture of **6** (12%), *cis*-**7** (22%), and *trans*-**7** (26%) together with some unreacted α-diazoamide (10%) (result not included in the table).

Interestingly, no product from the potentially competitive palladium-catalyzed cross-coupling of the aryl halide with the α-diazoamide moiety [32,33,34,35] was observed in the reactions of **5k** (Table 3, entries 41–42) and **5l** (Table 3, entries 45–46) when using Pd(0) catalysts.

As can be seen in Table 3, the effect of the substituent at the benzyl group on the outcome of the process varied according to its electronic nature as well as its position on the aromatic ring. The introduction of electron-donor groups led to an increased formation of the cycloheptapyrrolone product, especially when using Pd(0) catalysts. The increase was lower when the substituent was located at the *ortho*-position, probably due to steric interactions. In contrast, the electron-withdrawing groups generally diverted the palladacarbene away from the Buchner reaction in favor of the Csp^3^–H insertion. Similar electronic effects have been observed in related Rh(II)-catalyzed transformations [48]. On the other hand, the use of ((IMes)Pd(NQ))_2_ as the catalyst in the reaction of those substrates bearing electron-withdrawing groups resulted in the formation of minor amounts of the corresponding γ-lactam (**8**), which arises from the *^t^*butyl Csp^3^–H insertion. Notably, the γ-lactams were not observed when using Pd(II) catalysts.

The above results also confirm the significant impact of the electronic nature of the Pd catalyst and ensue electrophilicity of the carbene intermediate in the reaction pathway. Thus, whereas benzylic Csp^3^–H insertion is strongly favored over the Buchner reaction when using Pd(II), an increased cycloheptapyrrolone product formation is usually observed with the more electron-rich Pd(0) complexes. Interestingly, Rh(II)-catalyzed transformations show opposite reactivity trends, in which highly electrophilic Rh(II) complexes favor Buchner reactions over benzylic Csp^3^–H insertion [44,45,46].

Finally, we explored the transition metal-catalyzed decomposition of α-diazoacetamide **11**, which bears a (4-pyridinyl)methyl group instead of the *N*-benzyl substituent (Table 4). Treatment of **11** with a catalytic amount of ((IMes)Pd(NQ))_2_ under the optimized reaction conditions resulted in the complete decomposition of the material (Table 4, entry 1). When the reaction was performed at lower temperatures (refluxing dichloromethane), **11** was recovered unchanged (Table 4, entry 2). In contrast, the use of Pd(II) catalysts (Table 4, entries 3 and 4) resulted in the formation of β-lactam **12** (*cis*/*trans* mixture), with (SIPr)Pd(allyl)Cl affording the highest yield. Interestingly, α-diazoacetamide **11** was recovered unchanged when using the well-known (Rh(OAc)_2_)_2_ catalyst (Table 4, entry 5).

In our previous work on the synthesis of β-lactams, we studied computationally the Pd(II)-catalyzed insertion reaction of α-diazo-α-(methoxycarbonyl)acetamides [56]. According to our DFT calculations, this insertion reaction also occurs stepwise and involves an unprecedented Pd(II)-promoted Mannich-type reaction through a pallada(II)carbene-induced zwitterionic intermediate. To shed light on the reaction mechanism and the influence of the Pd(0) catalyst on the selectivity of the C–H insertion described above, DFT calculations were also carried out. To this end, the process involving **5a** in the presence of the model Pd(NHC) (NHC = 1,3-bis(phenyl)-imidazil-2-ylidene) catalyst was explored.

Figure 2 shows the computed reaction profile for the transformation of the initially formed pallada(0)carbene intermediate **INT0** into the observed β-lactams ***cis*-7a** and ***trans*-7a**. Similar to the reaction profile computed for the formation of oxindoles discussed above (Figure 1), the initial **INT0** readily evolves into the corresponding five-membered pallada(II)cycle **INT1** via the transition state **TS1** in a highly exergonic transformation (∆G_R_ ≈ –40 kcal/mol). Once again, this saddle point is associated with a 1,4-hydrogen migration reaction mediated by the transition metal fragment, thus confirming the generality of this type of C–H insertion reaction. Then, species **INT1** is transformed into the final β–lactam through a reductive elimination reaction via **TS2**, which releases the active catalyst able to enter into a new catalytic cycle. Interestingly, the computed *cis*/*trans* activation barrier differences for either the initial 1,4-H migration (∆G^≠^ = 1.3 kcal/mol) or for the subsequent elimination reaction (∆G^≠^ = 0.2 kcal/mol) indicates that the formation of the ***cis*-7a** β-lactam should be only slightly favored over the formation of its *trans* counterpart, which nicely matches the experimental findings (see Table 3, entry 2).

## 3. Materials and Methods

### 3.1. General Information

All commercially available reagents were used without further purification. ^1^H- and ^13^C-NMR spectra were recorded using Me_4_Si as the internal standard with a Varian Mercury 400 instrument (Oxford, United Kingdom). Chemical shifts are reported in ppm downfield (δ) from Me_4_Si for ^1^H and ^13^C-NMR. TLC was carried out on SiO_2_ (silica gel 60 F_254_, Merck), and the spots were located with UV light or 1% aqueous KMnO_4_. Flash chromatography was carried out on SiO_2_ (silica gel 60, SDS, 230–400 mesh ASTM). Organic extracts were dried over anhydrous Na_2_SO_4_ during workup of reactions. Evaporation of solvents was accomplished with a rotatory evaporator (Büchi, Flavill, Switzerland). α-Diazoacetamides **1a**–**1j** and oxindoles **2a**–**2j** are known compounds previously prepared by us [55], as are α-diazoacetamides **5a**–**5m** and β-lactams **7a**–**7m** [56].

### 3.2. Synthesis of N-Benzyl-N-(2-pyridinyl)-α-(ethoxycarbonyl)-α-diazoacetamide (**3**).

To a solution of 2-benzylaminopyridine (0.5 g, 2.71 mmol) and Et_3_N (0.39 mL, 2.71 mmol) in CH_2_Cl_2_ (10 mL), cooled at 0 °C, ethyl malonyl chloride (0.38 mL, 2.71 mmol) was added slowly. The mixture was stirred at room temperature for 24 h. After the reaction was completed, the mixture was poured into water and extracted with CH_2_Cl_2_. The organic extracts were washed with saturated NaHCO_3_ aqueous solution, dried, filtered, and concentrated. The residue was purified by chromatography (SiO_2_, from CH_2_Cl_2_ to CH_2_Cl_2_/MeOH 97:3) to give *N*-benzyl-*N*-(2-pyridinyl)-α-(ethoxycarbonyl) acetamide (0.77 g, 95%).

To a solution of *N*-benzyl-*N*-(2-pyridinyl)-α-(ethoxycarbonyl)acetamide (0.25 g, 0.84 mmol) and Et_3_N (0.14 mL, 1.0 mmol) in dry acetonitrile (15 mL), *p*-toluenesulfonylazide (2.6 mL of a 11% solution in toluene, 1.45 mmol) was added dropwise. The mixture was stirred at room temperature for 90 h. The solvent was removed in vacuo, and the resulting residue was partitioned between CH_2_Cl_2_ and 10% NaOH aqueous solution. The organic extracts were dried, filtered, and concentrated. The residue was purified by chromatography (SiO_2_, from CH_2_Cl_2_ to CH_2_Cl_2_/MeOH 99:1) to give *N*-benzyl-*N*-(2-pyridinyl)-α-(ethoxycarbonyl)-α-diazoacetamide (**3**, 150 mg; 55%) as a brown oil. ^1^H-NMR (CDCl_3_, 400 MHz) δ 1.07 (t, *J* = 7.2 Hz, 3H), 3.92 (q, *J* = 7.2 Hz, 2H), 5.22 (s, 2H), 7.04 (ddd, *J* = 7.4, 4.8, and 0.8 Hz, 1H), 7.10 (dt, *J* = 8.4 and 0.8 Hz, 1H), 7.20 (tt, *J* = 7.2 and 1.2 Hz, 1H), 7.23–7.29 (m, 2H), 7.32–7.36 (m, 2H), 7.60 (ddd, *J* = 8.4, 7.4 and 2.0 Hz, 1H), and 8.40 (ddd, *J* = 4.8, 2.0 and 0.8 Hz, 1H). ^13^C-NMR (CDCl_3_, 100.6 MHz) δ 14.3 (CH_3_), 52.3 (CH_2_), 61.4 (CH_2_), 70.2 (C), 118.3 (CH), 120.8 (CH), 127.4 (CH), 127.9 (2 CH), 128.6 (2 CH), 137.5 (C), 138.0 (CH), 148.6 (CH), 156.1 (C), 161.4 (C), and 162.3 (C).

### 3.3. Synthesis of N-tert-Butyl-N-(4-pyridinylmethyl)-α-(methoxycarbonyl)-α-diazoacetamide (**11**).

To a solution of *N*-tert-butyl-*N*-(4-pyridinylmethyl)amine (0.67 g, 4.1 mmol) and Et_3_N (0.58 mL, 4.1 mmol) in CH_2_Cl_2_ (20 mL), cooled at 0 °C, methyl malonyl chloride (0.57 mL, 5.3 mmol) was added slowly. The mixture was stirred at room temperature for 24 h. After the reaction was completed, the mixture was poured into water and extracted with CH_2_Cl_2_. The organic extracts were washed with saturated NaHCO_3_ aqueous solution, dried, filtered, and concentrated to give *N*-tert-butyl-*N*-(4-pyridinylmethyl)-α-(methoxycarbonyl)acetamide as an orange oil (0.98 g, 90%), which was used in the next reaction without purification.

To a solution of *N*-tert-butyl-*N*-(4-pyridinylmethyl)-α-(methoxycarbonyl)acetamide (0.5 g, 1.89 mmol) and DBU (0.45 mL, 2.85 mmol) in dry acetonitrile (6 mL), a solution of p-ABSA (515 mg, 2.1 mmol) in dry acetonitrile (2 mL) was added dropwise. The mixture was stirred at room temperature overnight. The solvent was removed in vacuo, and the resulting residue was partitioned between CH_2_Cl_2_ and 10% NaOH aqueous solution. The organic extracts were dried, filtered, and concentrated. The residue was purified by chromatography (SiO_2_, from CH_2_Cl_2_ to CH_2_Cl_2_/MeOH 98:2) to give *N*-tert-butyl-*N*-(4-pyridinylmethyl)-α-(methoxycarbonyl)-α-diazoacetamide (**11**, 145 mg; 26%) as an orange oil. ^1^H-NMR (CDCl_3_, 400 MHz) δ 1.40 (s, 9H), 3.77 (s, 3H), 4.63 (s, 2H), 7.15 (d, *J* = 6.0 Hz, 2H), and 7.57 (d, *J* = 6.0 Hz, 2H). ^13^C-NMR (CDCl_3_, 100.6 MHz) δ 29.0 (3 CH_3_), 50.6 (CH_2_), 52.4 (CH_3_), 59.5 (C), 68.7 (C), 121.9 (2 CH), 149.3 (C), 150.2 (2 CH), 162.8 (C), and 163.5 (C).

### 3.4. Characterization Data for New Compounds of Scheme 4 and Tables 3 and 4

*1-Benzyl-3-(ethoxycarbonyl)-2-oxo-2,3-dihydro-1H-imidazo (1,2-a)pyridin-4-ium-3-ylide* (**4**). Amorphous orange solid. ^1^H-NMR (CDCl_3_, 400 MHz) δ 1.45(t, *J* = 7.2 Hz, 3H), 4.43 (q, *J* = 7.2 Hz, 2H), 5.20 (s, 2H), 6.99 (ddd, *J* = 8.8, 1.6, and 0.8 Hz, 1H), 7.08 (ddd, *J* = 7.6, 6.8, and 1.2 Hz, 1H), 7.25–7.37 (m, 5H), 7.40 (ddd, *J* = 8.8, 7.6, and 1.2 Hz, 1H), 9.65 (d, *J* = 6.8 Hz, 1H). ^13^C-NMR (CDCl_3_, 100.6 MHz) δ 14.9 (CH_3_), 43.3 (CH_2_), 60.0 (CH_2_), 94.0 (C), 106.4 (CH), 115.7 (CH), 127.8 (2 CH), 128.2 (CH), 129.0 (CH), 129.1 (2 CH), 129.9 (CH), 135.4 (C), 135.6 (C), 157.0 (C), and 162.6 (C).

*Methyl 2-tert-butyl-6-chloro-3-oxo-1H-2,3-dihydrocyclohepta(c)pyrrole-3a-carboxylate* (**6d**). ^1^H-NMR (CDCl_3_, 400 MHz, signals from a 4.5:1 mixture of **trans-7d** and **6d**) δ 1.45 (s, 9H), 3.64 (s, 3H), 4.21 (dd, *J* = 14.8 and 1.2 Hz, 1H), 4.44 (d, *J* = 14.8 Hz, 1H), 5.63 (d, *J* = 10.2 Hz, 1H), 6.17 (d, *J* = 6.8 Hz, 1H), 6.39 (d, *J* = 10.2 Hz, 1H), and 6.62 (d, *J* = 6.8 Hz, 1H).

*Methyl 1-(4-cyanobenzyl)-5,5-dimethyl-2-oxopyrrolidine-3-carboxylate* (**8e**). Amorphous orange solid. ^1^H-NMR (CDCl_3_, 400 MHz) δ 1.16 (s, 3H), 1.22 (s, 3H), 2.19 (dd, *J* = 13.2 and 9.2 Hz, 1H), 2.33 (dd, *J* = 13.2 and 9.2 Hz, 1H), 3.62 (t, *J* = 9.2 Hz, 1H), 3.82 (s, 3H), 4.33 (d, *J* = 16.0 Hz, 1H), 4.60 (d, *J* = 16.0 Hz, 1H), 7.39 (d, *J* = 8.4 Hz, 2H), 7.60 (d, *J* = 8.4 Hz, 2H). ^13^C-NMR (CDCl_3_, 100.6 MHz) δ 27.1 (CH_3_), 28.1 (s, CH_3_), 38.3 (CH_2_), 43.2 (CH_2_), 47.4 (CH), 53.0 (CH_3_), 60.0 (C), 111.5 (C), 118.8 (C), 128.3 (2 CH), 132.6 (2 CH), 143.9 (C), 170.2 (C), and 170.9 (C).

*Methyl cis-1-tert-butyl-4-[4-(dimethylamino)phenyl]-2-oxoazetidine-3-carboxylate* (cis**-7f**). Amorphous orange solid. ^1^H-NMR (CDCl_3_, 400 MHz) δ 1.30 (s, 9H), 2.95 (s, 6H), 3.40 (s, 3H), 4.18 (d, *J* = 6.0 Hz, 1H), 4.83 (d, *J* = 6.0 Hz, 1H), 6.65 (d, *J* = 8.8 Hz, 2H), and 7.21 (d, *J* = 8.8 Hz, 2H).

*Methyl 2-tert-butyl-7-chloro-3-oxo-1H-2,3-dihydrocyclohepta[c]pyrrole-3a-carboxylate (**6g**).*^1^H-NMR (CDCl_3_, 400 MHz, signals from a 11:1 mixture of *trans*-**7g** and **6g**) δ 1.45 (s, 9H), 3.64 (s, 3H), 4.22 (dd, *J* = 15.6 and 2.0 Hz, 1H), 4.47 (dd, *J* = 15.6 and 2.4 Hz, 1H), 5.60 (d, *J* = 9.6 Hz, 1H), 6.27–6.30 (m, 1H), 6.32 (dd, *J* = 9.6 and 7.2 Hz, 1H), and 6.61 (dd, *J* = 7.2 and 1.2 Hz, 1H).

*Methyl 2-tert-butyl-5-chloro-3-oxo-1H-2,3-dihydrocyclohepta[c]pyrrole-3a-carboxylate (**6g’**).*^1^H-NMR (CDCl_3_, 400 MHz, signals from a 12:4:1 mixture of *trans*-**7g**, **6g**, and **6g’**) δ 1.46 (s, 9H), 3.64 (s, 3H), 4.23 (dd, *J* = 15.2 and 2.0 Hz, 1H), 4.42 (dd, *J* = 15.2 and 2.4 Hz, 1H), 5.76 (t, *J* = 1.2 Hz, 1H), 6.18–6.21 (m, 1H), and 6.38–6.40 (m, 2H).

*Methyl 1-(3-cyanobenzyl)-5,5-dimethyl-2-oxopyrrolidine-3-carboxylate (**8h**).*^1^H-NMR (CDCl_3_, 400 MHz, significant signals from a 2:1 mixture of **8h** and *cis*-**7h**) δ 1.16 (s, 3H), 1.24 (s, 3H), 2.19 (dd, *J* = 12.8 and 9.2 Hz, 1H), 2.33 (dd, *J* = 12.8 and 9.2 Hz, 1H), 3.62 (t, *J* = 9.2 Hz, 1H), 3.80 (s, 3H), 4.33 (d, *J* = 16.0 Hz, 1H), and 4.55 (d, *J* = 16.0 Hz, 1H).

*Methyl 2-tert-butyl-8-methoxy-3-oxo-1H-2,3-dihydrocyclohepta[c]pyrrole-3a-carboxylate (**6i**).*^1^H-NMR (CDCl_3_, 400 MHz, signals from a 1.5:1 mixture of **6i** and *cis*-**7i**) δ 1.47 (s, 9H), 3.55 (s, 3H), 3.69 (s, 3H), 4.23 (dd, *J* = 14.4 and 1.6 Hz, 1H), 4.48 (dd, *J* = 14.4 and 1.6 Hz, 1H), 5.65 (d, *J* = 7.2 Hz, 1H), 6.29–6.33 (m, 1H), 6.31 (d, *J* = 7.6 Hz, 1H), and 6.38–6.45 (m, 1H). ^13^C-NMR (CDCl_3_, 100.6 MHz, significant signals from a 1:8 mixture of **6i** and ***cis*-7i**) δ 27.6 (3 CH_3_), 49.4 (CH_2_), 52.8 (CH_3_), 55.1 (C), 57.5 (CH_3_), 99.8 (CH), 120.6 (CH), 122.5 (CH), and 126.9 (CH).

*Methyl 1-(2-fluorobenzyl)-5,5-dimethyl-2-oxopyrrolidine-3-carboxylate (**8j**).*^1^H-NMR (CDCl_3_, 400 MHz, signals from a 8:1 mixture of **8j** and *cis*-**7j**) δ 1.14 (s, 3H), 1.25 (s, 3H), 2.15 (dd, *J* = 12.8 and 9.2 Hz, 1H), 2.31 (dd, *J* = 12.8 and 9.2 Hz, 1H), 3.61 (t, *J* = 9.2 Hz, 1H), 3.82 (s, 3H), 4.46 (d, *J* = 16.0 Hz, 1H), 4.55 (d, *J* = 16.0 Hz, 1H), 7.00 (ddd, *J* = 10.4, 8.4, and 1.2 Hz, 1H), 7.09 (td, *J* = 7.6 and 1.2 Hz, 1H), 7.19–7.25 (m, 1H), and 7.39 (td, *J* = 7.6 and 1.2 Hz, 1H).

*Methyl 1-(2-bromobenzyl)-5,5-dimethyl-2-oxopyrrolidine-3-carboxylate (**8k**).* Amorphous orange solid. ^1^H-NMR (CDCl_3_, 400 MHz) δ 1.18 (s, 3H), 1.23 (s, 3H), 2.20 (dd, *J* = 12.8 and 9.2 Hz, 1H), 2.35 (dd, *J* = 12.8 and 9.2 Hz, 1H), 3.65 (t, *J* = 9.2 Hz, 1H), 3.83 (s, 3H), 4.45 (d, *J* = 16.4 Hz, 1H), 4.64 (d, *J* = 16.4 Hz, 1H), 7.10 (ddd, *J* = 8.0, 6.8 and 2.4 Hz, 1H), 7.24–7.30 (m, 2H), 7.51 (dd, *J* = 7.6 and 1.2 Hz, 1H). ^13^C-NMR (CDCl_3_, 100.6 MHz) δ 26.9 (s, CH_3_), 27.8 (s, CH_3_), 38.3 (CH_2_), 43.1 (CH_2_), 47.6 (CH), 52.9 (CH_3_), 60.0 (C), 122.6 (C), 128.0 (CH), 128.9 (CH), 129.2 (CH), 132.7 (CH), 137.1 (C), 170.1 (C), and 171.1 (C).

*Methyl 1-(2-iodobenzyl)-5,5-dimethyl-2-oxopyrrolidine-3-carboxylate (**8l**).* Amorphous orange solid. ^1^H-NMR (CDCl_3_, 400 MHz) δ 1.20 (s, 3H), 1.22 (s, 3H), 2.20 (dd, *J* = 13.2 and 9.2 Hz, 1H), 2.35 (dd, *J* = 13.2 and 9.2 Hz, 1H), 3.65 (t, *J* = 9.2 Hz, 1H), 3.83 (s, 3H), 4.35 (d, *J* = 16.4 Hz, 1H), 4.59 (d, *J* = 16.4 Hz, 1H), 6.93 (td, *J* = 8.0 and 1.6 Hz, 1H), 7.23 (dd, *J* = 8.0 and 1.6 Hz, 1H), 7.30 (td, *J* = 8.0 and 1.2 Hz, 1H), and 7.79 (dd, *J* = 8.0 and 1.2 Hz, 1H).

*Methyl cis-1-tert-butyl-2-oxo-4-(pyridin-4-yl)azetidine-3-carboxylate (**cis-12**).*^1^H-NMR (CDCl_3_, 400 MHz, signals from a 2.6:1 mixture of *trans*-**12** and *cis*-**12**) δ 1.32 (s, 9H), 3.36 (s, 3H), 4.28 (d, *J* = 6.4 Hz, 1H), 4.88 (d, *J* = 6.4 Hz, 1H), 7.32-7.35 (m, 2H), and 8.61–8.65 (m, 2H). ^13^C-NMR (CDCl_3_, 100.6 MHz, signals from a 2.6:1 mixture of ***trans*-12** and ***cis*-12**) δ 28.2 (3 CH_3_), 52.2 (CH_3_), 55.4 (CH), 55.5 (C), 59.0 (CH), 121.1 (2 CH), 146.2 (C), 150.2 (2 CH), 162.5 (C), and 165.9 (C).

*Methyl trans-1-tert-butyl-2-oxo-4-(pyridin-4-yl)azetidine-3-carboxylate (trans-**12**).*^1^H-NMR (CDCl_3_, 400 MHz) δ 1.27 (s, 9H), 3.67 (d, *J* = 2.4 Hz, 1H), 3.78 (s, 3H), 4.84 (d, *J* = 2.4 Hz, 1H), 7.33 (dd, *J* = 4.4 and 1.6 Hz, 2H), and 8.64 (dd, *J* = 4.4 and 1.6 Hz, 2H). ^13^C-NMR (CDCl_3_, 100.6 MHz) δ 28.2 (3 CH_3_), 53.0 (CH_3_), 55.2 (CH), 55.7 (C), 62.1 (CH), 121.6 (2 CH), 148.5 (C), 150.7 (2 CH), 161.7 (C), and 167.0 (C).

## 4. Computational Details

All the calculations reported in this paper were performed with the Gaussian 09 suite of programs [66]. Electron correlation was partially taken into account using the hybrid functional usually denoted as B3LYP [67,68,69] in conjunction with the D3 dispersion correction suggested by Grimme et al. [70] using the standard double-ζ quality def2-SVP [71,72] basis set for all atoms. The Polarizable Continuum Model (PCM) [73,74,75] was used to model the effects of the solvent. This level is denoted PCM(solvent)-B3LYP-D3/def2-SVP. Geometries were fully optimized in solution without any geometry or symmetry constraints. Reactants, intermediates, and products were characterized by frequency calculations and have positive definite Hessian matrices. Transition structures (TSs) show only one negative eigenvalue in their diagonalized force constant matrices, and their associated eigenvectors were confirmed to correspond to the motion along the reaction coordinate under consideration using the Intrinsic Reaction Coordinate (IRC) method [76]. Frequency calculations were also used to determine the difference between the potential (E) and Gibbs (G) energies, G − E, which contains the zero-point, thermal, and entropy energies. Potential energies were refined, E_sol_, by means of single point (SP) calculations at the same level with a larger basis set, def2-TZVPP [71,72], where all elements were described with a triple-ζ plus polarization quality basis set. This level is denoted PCM(solvent)-B3LYP-D3/def2-TZVPP//PCM(solvent)-B3LYP-D3/def2-SVP. The ΔG and ΔG^≠^ values given in the text were obtained from the Gibbs energy in solution, G_sol_, which was calculated by adding the thermochemistry corrections, G − E, to the refined SP energies, E_sol_, i.e., G_sol_ = E_sol_ + G − E. See Appendix A.

## 5. Conclusions

In summary, in the present paper, we report a full account of our experimental and computational studies on the Pd-catalyzed intramolecular carbene C–H insertion of α-diazo-α-(methoxycarbonyl)acetamides to prepare oxindoles and β-lactams. We have explored how the reactions are affected by the substituents on the α-diazoamide moiety and by the catalyst type, exploring the use of both Pd(0) and Pd(II) catalysts.

The chemoselectivity of the Pd-catalyzed C–H insertion reactions of α-diazo acetamides is mainly governed by the nature of the substrates. Thus, while *N*-aryl acetamides selectively afforded oxindoles, starting from *N*,*N*-dialkyl acetamides, the corresponding β-lactams were obtained.

The results obtained in the annulation reactions to form oxindoles show that Pd(0) catalysts were much more efficient than Pd(II) catalysts and that ((IMes)Pd(NQ))_2_ can be used as an alternative to Pd_2_(dba)_3_ to catalyze the carbene C_Ar_sp^2^–H insertion.

On the other hand, both Pd(0) and Pd(II) catalysts can be used to promote Csp^3^–H insertion leading to β-lactams. However, the Pd(II) complexes proved to be more versatile for β-lactam formation since the use of Pd(0) catalysts resulted in increased amounts of Buchner products and, in some cases, of the corresponding γ-lactams.

Our computational studies show that when using ((IMes)Pd(NQ))_2_ as the catalyst, both C_Ar_sp^2^–H and Csp^3^–H insertions involve similar stepwise reaction mechanisms, in which a palladium-mediated hydrogen migration is followed by a reductive elimination.

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
