# Peer review of "Site Selectivity in Pd-Catalyzed Reactions of α-Diazo-α-(methoxycarbonyl)acetamides: Effects of Catalysts and Substrate Substitution in the Synthesis of Oxindoles and β-Lactams"

_molecules, 2019, doi:10.3390/molecules24193551_

Round 1

Reviewer 1 Report

This is a nice paper, which reports a full account on the intramolecular insertion of a-diazo-(methoxycarbonyl)acetamides to give oxindoles or beta-lactams using different Pd(0) and Pd(II) catalysts. The manuscript has been well written and the reports results (both experimental and computational) are interesting and deserve to be published in Molecules, after the following minor revisions:

1) In the Introduction, some comments on the already published palladium-catalyzed procedures leading to oxindoles or beta-lactams functionalized with the alkoxycarbonyl group should be given (and the related articles cited in the reference list). See, in particular, the Pd(II) catalyzed carbonylative processes published in Tetrahedron Lett. 1995, 36, 7495 (beta-lactams); Eur. J. Org. Chem. 2001, 4607 (oxindoles)

2) In the Results and Discussion section, the authors should better point out that, under their conditions, the selectivity toward the formation of either the oxindole (Table 2) or the beta-lactam (Table 4) derivatives is governed by the nature of the substrate (1 or 5, respectively).

Author Response

Reviewer 1:

We appreciate the comments by this reviewer.

Following the recommendation of the reviewer, some new sentences and references (57-60 and 63-65) have been incorporated in the manuscript to account for either transition metal-catalyzed or metal-free procedures in the synthesis of oxindoles and β-lactams:

“The aim of the current work was to gain more insight into the Pd-catalyzed carbene insertion reactions of α-diazo-α-(methoxycarbonyl)acetamides leading to oxindoles and β-lactams. In order to ascertain the factors governing the selectivity of the processes, . To this end, we explored how the reactions are affected by the substituents on the α-diazoacetamide and the type of catalyst, using complexes with two oxidation states of Pd and a variety of ligands (Scheme 3).”

“The oxindole system is a common structural motif in pharmaceuticals and natural products, and the development of efficient procedures for their preparation is therefore of considerable interest [57-60].”

“Due to its ubiquitous presence in the molecular structure of natural products and biologically active compounds, β-lactam synthesis has attracted considerable attention over the years [63-65].”

A new sentence has been incorporated in the Conclusions section to further clarify the control of the selectivity in the palladium-catalyzed reactions:

“The chemoselectivity of the Pd-catalyzed C–H insertion reactions of α-diazo acetamides is mainly governed by the nature of the substrates. Thus, while N-aryl acetamides selectively afforded oxindoles, starting from N,N-dialkyl acetamides the corresponding β-lactams were obtained.”

Reviewer 2 Report

The Authors report the use of [(IMes)Pd(NQ)]2 as an alternative to Pd2(dba)3 for the Pd-catalyzed intramolecular carbene C–H insertion of α-diazo-α-(methoxycarbonyl)acetamides to prepare oxindoles and β-lactams. The results show that, in the preparation oxindoles, the Pd(0)-catalysts were much more efficient than Pd(II)-catalysts and that [(IMes)Pd(NQ)]2 can be used as an alternative to Pd2(dba)3 to catalyze the carbene CArsp2–H insertion, while Pd(II)-complexes proved to be more versatile to promote Csp3–H insertion in the preparation of β-lactams.

DFT calculation of pallada(0)carbene into oxindoles and β-lactams has been performed.

The paper is well written, the results clearly presented and the conclusions supported by the results.

The paper can be accepted for publication in the present form.

Author Response

Reviewer 2:

We appreciate the positive comments by this reviewer.